# Comparative Study of Comfort Indicators for School Constructions in Sustainability Methodologies: Schools in the Amazon and the Southeast Region of Brazil

**Tatiana Santos Saraiva** [1,*] **, Edson Martins da Silva** [2] **, Manuela Almeida** [3] **and Luís Bragança** [3]

[1] International Doctoral Program in Sustainable Built Environment, School of Engineering, Universidade do Minho, 4800058 Guimarães, Portugal

[2] Department of Architecture, CEAP—Centro de Ensino Superior do Amapá, Macapá 68906-646, Brazil; edsontstcdo@gmail.com

[3] Department of Civil Engineering, School of Engineering, Universidade do Minho, 4800058 Guimarães, Portugal; malmeida@civil.uminho.pt (M.A.); braganca@civil.uminho.pt (L.B.)

* Correspondence: saraivaus@yahoo.com; Tel.: +55-96-98106-1627

**Abstract:** In the 1970s and 1980s, the effects of pollution in the atmosphere grew from a local to a global scale, affecting the entire planet. Consequently, economic and social issues began to be increasingly more connected with environmental protection. Several measures were taken towards environmental preservation, many of those related to the sustainable construction of buildings. This work intended to make a parallel between comfort indicators among different schools in Brazil, beginning with the specific analyses of schools in Juiz de Fora (Minas Gerais, MG), in the Southeast region, and in Macapá (Amapa, AP), in the Amazon or Northern region. This comparison between schools is made to demonstrate the need to adapt methodologies and certifications that promote sustainability in school buildings for each region of Brazil. Questionnaires about ergonomic, thermal, visual and acoustic comfort, and air quality, were applied in two high-school buildings in Juiz de Fora, Academia School and Santa Catarina School, and in two high schools in Macapá, Tiradentes School and Professor Gabriel Almeida Café school, to understand the difference between these two regions of Brazil regarding comfort conditions. A comparison between the results of the four schools was made, proving the necessity of the elaboration of a specific methodology for each Brazilian region.

**Keywords:** comfort indicators; Amapá; Juiz de Fora; sustainability in school constructions

## 1. Introduction

The environmental impact produced by construction activities has enlarged over the past few decades [1]. Therefore, in an attempt to solve this, some initiatives started to appear, such as methodologies to evaluate and certificate the sustainability of the buildings. Over the years, progress has been made and several BSA (building sustainability assessment) methodologies were created. These tools are adapted according to the necessity of the specific region in question [2] and are intended to support a decrease in the overall environmental impact.

Nowadays, there is no specific federal program related to the architecture of schools, but rather specific legislation for sizing, number of students and climate, among other factors. School buildings are constructed with respect to the specific requirements of each region. Many systems for the assessment of buildings in the field of sustainability can be found in the literature, having been created as a way to help mitigate the consequences of the world energy crisis in the decade of the 1970s.

Some attained greater international prominence, namely LEED (Leadership in Energy and Environmental Design), which was developed in the United States; BREEAM (Building Research Establishment Environmental Assessment Methodology) in the United Kingdom; HQE (Haute Qualité Environnementale) in France; and SBTool (Sustainable Building Tool, iiSBE), developed by a group of experts from twenty-two countries belonging to the iiSBE, the International Initiative for a Sustainable Built Environment [3].

The relevance of this work is related to the comparison, regarding the environmental comfort of classrooms, between schools from two Brazilian regions, the Amazon region (Macapá) and the Southeast region (Juiz de Fora), with very diverse characteristics related to social, cultural, economic and climatic aspects, demonstrating the great need to adapt the sustainable methodologies to the specificities of each region.

Sustainable construction pursues a balance among three aspects: social, environmental and economic. Environmental comfort, in sustainable methodologies, belongs to the social aspect. The environmental comfort of building users should be ensured through passive solutions, such as the proper use of natural lighting and ventilation, seeking energy efficiency, thereby reducing costs and the environmental impact of the building.

This article is part of the author's thesis, which aims to adapt the SBTool$^{PT}$ methodology for high schools according to the Portuguese reality. This thesis also wants to explore the possibility of expanding the SBTool$^{PT}$ methodology to the Brazilian context and this paper has as its main objective to start research on the adaptation of sustainability assessment tools in Brazil, with a special emphasis on the development of indicators related to environmental comfort. The results obtained through the application of surveys to students from the city of Juiz de Fora are evaluated [4] together with those from the city of Macapá. Through this comparison, the need to adapt sustainability methodologies, such as SBTool, LEED and BREEAM, among others, to the specific reality of each Brazilian region is analyzed. In the following section, the main aspects of the regions of Brazil are shown.

## 2. Overview of Brazilian Regions

Brazil is a country with huge dimensions, with a significant diversity of cultures, economies and climates. It is the fifth biggest country in the world, with an area of 8,515,767 km$^2$. Brazil is divided into five regions: Southeast, South, Centre-West, Northeast and North. There are several types of climatic regions in Brazil, such as subtropical, semiarid, Atlantic tropical, high-altitude tropical, tropical and equatorial.

The flora in Brazil has a great diversity, such as tropical rain forest, savanna, thorny shrub, periodically wet land, tropical semi-deciduous forest and grassland. The GDP per capita differs for each region, with Brasilia having the largest, followed by Sao Paulo. The states with the lowest GDP per capita are in the north and northeast. All these aspects interfere in the selection of construction practices and also in the types of buildings [5,6].

The main characteristics of the Brazilian regions are described in the following paragraphs [7]:

(1). South Region:

- Area: 576,774 km$^2$, about 7% of the total area, has a high population density (about 43.50 inhabitants/km$^2$) and 14.36% of the Brazilian population [6].
- Formed by the following states: Paraná, Santa Catarina and Rio Grande do Sul.
- Climate: subtropical, lowest temperatures in the country, with well-defined seasons of the year and rains distributed regularly throughout the year.
- Socioeconomic aspects: high social, economic and cultural development, similar across all sectors (education, health, economy and others).

(2). Southeast Region:

- Area: 924,620 km$^2$, being the second smallest region of the country, but it has the highest population density (about 85 inhabitants/km2), about 43% of the Brazilian population [6].
- Formed by the following states: São Paulo, Minas Gerais, Rio de Janeiro and Espírito Santo.
- Climate: tropical, oscillating between temperate and hot, with large local variations and two well defined seasons, one rainy (summer) and one dry (winter).
- Socioeconomic aspects: very urbanized (91% of the inhabitants live in urban areas). The region is the second in quality of life, has the highest GDP per capita, the most satisfactory water supply system in the country and the best offer of basic health services, is an important industrial, commercial and financial region of the country employing 70% of Brazilian workers and using around 85% of the total electricity consumed in the country.

(3). Central-West Region:

- Area: 1,606,403 km2, 18.86% of the national territory, the second largest region of the country in territorial area, with the second lowest population density, with 7.36% of the total population of the country, showing large demographic gaps [6].
- Formed by the following states: Goiás, Mato Grosso, Mato Grosso do Sul and the Federal District, where the capital of the country, Brasília, is located.
- Climate: tropical, hot and rainy, with hot summer seasons and dry winters.
- Socioeconomic aspects: serious deficiencies in the water supply system in the rural area. This region is in a constant process of development, having numerous incentives and investments in transport infrastructures, which contributed to the growth and modernization of the region.

(4). Northeast Region:

- Area: 1,554,291 km$^2$, 18% of the national territory, with 27.83% of the Brazilian population [6].
- Formed by the following states: Alagoas, Bahia, Ceará, Maranhão, Paraíba, Piauí, Pernambuco, Rio Grande do Norte and Sergipe.
- Climate: tropical (Bahia, Ceará, Maranhão and Piauí) and Semi-arid (northeast of the interior of the region), due to the low average of annual rainfall.
- Socioeconomic aspects: it has varying levels of human development throughout its geographical zones, having the lowest average index in the country. The socioeconomic deficiencies in the development of this region are the lack of basic health, education and housing services.

(5). North region:

- Area: 3,853,676 km$^2$, 45.27% of the national territory, being the most extensive region of Brazil, and the least populated in the country, with a population density of only 4.77 inhabitants per km$^2$, corresponding to 8.32% of the population of the country [6].
- Formed by the following states: Acre, Amapá, Amazonas, Pará, Rondônia, Roraima and Tocantins.
- Climate: the humid equatorial climate provides the region an elevated temperature throughout the year and with low thermal amplitude.
- Socioeconomic aspects: many socio-environmental problems, such as severe deficiencies in access to water supply in urban areas, provision of basic health services, adequate housing, among others.

Given the above, it can be seen that there are great differences between each region in economic, social, political, cultural and climatic aspects.

## 3. Literature Review

Environmental comfort is a popular theme in all assessment schemes aimed at increasing the safety, health and comfort of the occupants of a building [7]. In most sustainability methodologies

for school buildings, such as LEED BD + C Schools, SBTool International for School and BREEAM Education, the most common comfort indicators are visual, acoustic, lighting and thermal comfort, as well as air quality [8].

It is important to add the ergonomic comfort indicator to the methodologies that assess sustainability in schools, since students spend at least 5 hours a day sitting in school chairs [9]. Therefore, the comfort related to the space and the furniture suited to the student's needs is extremely important. Subsequent sections present a literature review of each environmental comfort indicator carried out in order to acquire more knowledge on the subject.

### 3.1. Indoor Air Quality

Indoor air quality (IAQ) is determined by the building's capability to preserve the well-being and health of its users. Adequate air quality, temperature and humidity are essential for the health and interior comfort of building users [10].

Natural ventilation is one of the most economical and simple ways of reaching interior thermal comfort and is an effective tool for eliminating or reducing indoor pollutants emitted from internal sources, thereby improving the quality of indoor air. The purpose of ventilation is to provide suitable air renewal inside a building environment to achieve good indoor air quality. Artificial or natural ventilation can be used in accordance with the specific needs of each environment [10].

Three processes should be considered for proper management of the IAQ: emission source control, measurement of indoor air and verification of ventilation. These three processes occur in different phases of the lifecycle of a building. Emission source control happens through the choice of materials for the internal environment. [11].

Measurement of indoor air quality occurs before or after the occupation of a building, and measured pollutants include particulate and chemical pollutants. Laws and regulations on the comfort of the indoor environment vary between countries due to different economic, environmental, historical and political contexts [11].

Carbon dioxide is one of the main environmental factors that needs to be controlled, and that can be achieved through appropriate ventilation [12]. In recent decades, several policies and guidelines were developed in order to help designers build properly ventilated buildings. In the case of mechanically ventilated buildings, there are several standards, national and international, which indicate the minimum ventilation rates required for a satisfactory quality of indoor air.

Inadequate air quality conditions in learning spaces can cause reduction in student performance of up to 30%. High $CO_2$ concentrations are a result of poor ventilation and these low ventilation rates hinder teaching and learning by reducing the concentration and memory of students. The air-conditioning system helps to generate proper thermal comfort, but also reduces the relative humidity of the indoor air [13].

### 3.2. Indicator of Thermal Comfort

The environmental conditions necessary for comfort are not the same for everyone. "An environment must provide thermal conditions so that at least 80% of the occupants are satisfied with the thermal environment" [14].

Human thermal comfort can be addressed with two approaches, the adaptive model and the classical model. The adaptive model relates acceptable temperature ranges or indoor design temperatures to external climatological or meteorological parameters (outdoor air temperature). The classical model, also designated Fanger's model, takes into account that human comfort depends on six parameters: clothing insulation, temperature, metabolic rate, mean radiant temperature, air velocity and water vapor pressure [15]. In this way, regarding the environmental conditions that are required for thermal comfort in the classroom, they vary from person to person as a result of the wide variations between people [16].

An example of bioclimatic evaluation methods is the Bioclimatic Letters of Olgyay and Givoni, which are graphical representations of the relationship between climate and thermal comfort. This work aims to connect visual variables such as design strategies, physiological patterns of thermal comfort and weather conditions. Bioclimatic charts, which were the first thermal-comfort diagrams, combined dry bulb temperature and relative humidity, determining comfort zones and showing how these areas can be transformed in the presence of sunlight and ventilation [17]. Thermal comfort is related to the learning progress, and sun heat is a problem where external shading is absent [18].

### 3.3. Visual Comfort

The penetration of daylight into a building is important to ensure good performance and visual comfort for building occupants. This can be achieved by providing sufficient illumination levels and good daylight illumination, controlling the intense glare of the sun through the shape of the building, using efficient lighting fittings to reduce electricity consumption and allowing occupants to control lighting [8].

Five parameters affect lighting performance in the built environment, namely the external obstructions, type of glass, area and orientation of the building, shading and window area. Other factors that interfere with the process of luminal comfort are the characteristics of the internal environment, that is, its dimensions and colors (walls, ceilings, floors and furniture) [19].

Light has the main impact on learning progress when compared to other design parameters. Nevertheless, the size of the windows alone was not meaningfully correlated with learning progress, only when there was concern about both the risk of glare and the orientation of the window could students benefit from the ideal window size. Both the quality and quantity of electrical lighting have a significant relation with students' learning progress. Good visual comfort in the learning place aims to reduce the harm to the eye and the possibility of accidents with extreme visual accuracy [18].

### 3.4. Acoustic Comfort

Noise is a non-articulated sound that, according to its frequency and intensity, can cause discomfort and, in some cases, affect health [20].

Noise prevention and control in school buildings begins in project design, including definition of use, land use and also the choice of building materials appropriate for each environment. A project without major acoustic concerns causes difficulties in teaching and learning and may cause the need for repetition, interruption in explanations and elevation of the natural level of the voice. It is necessary to isolate zones that produce excessive noise, such as music rooms and sport fields, to protect the heating and ventilation ducts and to design correct geometry for the classrooms, being also important the use of adequate materials that prevent reverberation and with good insulation properties [21].

The design of the classroom should minimize the number of windows and doors directed to main roads, so as not to harm the requirements of thermal and lighting comfort [22].

There is some evidence to support the relationship between some design strategies, such as carpet area and room shape, and the reverberation time. These factors were significantly correlated with the learning rate. In addition, external and internal noise had a substantial negative impact on performance. Factors that affect the noise level, such as busy areas adjacent to the classrooms and the distance from the main traffic, show a correlation with the learning rate [18].

There is a huge need to develop techniques that provide adequate acoustic comfort conditions for building users, as well as a need to raise awareness among architects and designers about the design and dimensioning of human space to create an adequate acoustic environment for their activities.

### 3.5. Ergonomic Comfort

Ergonomics is the study of the adaptation of work to humans, involving organizational aspects and the physical environment related to the activities performed on site. In addition, ergonomics studies several aspects of human behavior.

Anthropometric measurements should be considered in the design of school furniture. For the proper design of classroom furniture, it is necessary to collect some specific anthropometric measurements such as thigh height, popliteal height, lumbar and elbow support space. Moreover, the height of the lumbar support is required to determine the dimensions of the furniture, which are important for achieving a proper posture. Classroom furniture is used continuously, so it is important that the products are of good quality in order to withstand frequent use [23].

In 1979, a general rule was developed with guidelines for the ergonomic design of classroom furniture: ISO 5970 (Standards for tables and chairs for educational institutions). This highlighted the importance of effectively obtaining anthropometric measurements for children and adolescents. Anthropometric measurements of high-school students vary among several ethnic groups, genders, ages, races and cultures. Consequently, it is difficult for a designer to define specific measurements for a different number of students [24].

## 4. Materials and Methods

This work intended to study the level of satisfaction within the school environment, in the cities of Macapá (Amapa) and Juiz de Fora (Minas Gerais).

The activity times in high schools usually occur in three shifts: morning (7 am to 12 pm), afternoon (1 pm to 6 pm) and night (7 pm to 11 pm). The minimum number of annual hours defined by the Law of Directives and Bases of National Education is 800 hours per year, spread over a minimum of 200 school days. Through these data, it can be concluded that a student stays in school around 5 hours a day.

This study sought to evaluate two high-school buildings. In Brazil, high school is the second cycle of basic education, attended by children aged 14–18 years old (10th to 12th grade).

This article continues the study by Saraiva et al. [4], in which it is made a parallel among the sustainable aspects of schools in Guimarães (Portugal), and Juiz de Fora (Brazil). Therefore, the questionnaire in this work is similar to the one used in that article [4], which was applied to students in Guimarães, Portugal.

The multiple-choice questionnaire was applied to school students from Macapá and Juiz de Fora, with the objective of identifying the level of satisfaction of students with the comfort inside the learning spaces. The responses of the survey applied to students in those high schools were divided into 5 levels: comfortable, slightly uncomfortable, uncomfortable and very uncomfortable.

The performance of the responses gave by the students was made by a statistical analysis applied by Microsoft Excel, considering 0.05 as a level of probability (reliability coefficient). The results determine the comfort level of the students inside the classroom of the analyzed schools. The questionnaire has six questions (Appendix A), whose options range from high discomfort to comfort [4].

The surveys were made in the winter, when the values of the temperatures recorded were 8–15 °C in the city of Juiz de Fora and 25–33 °C in Macapá. The recorded air humidity was about 80% in the cities. This questionnaire consists of six questions related to comfort environments. Through the results of the questionnaires, using the Microsoft Excel software ANOVA, an analysis was undertaken testifying to what level the average of each measured variable was correlated to the global average.

The analysis of variance (ANOVA), in mathematical terms, consists of the calculation of the value of F by the division between median squares of the model, Ftabulated, and the residual median squares, Fcalculated. ANOVA is a system used to check the equivalence of medians from several sets. The adequate result is reached when F is higher, representing that the median squares of the model are superior than the residual mediam squares, thus demonstrating that there is a variance among these sets.

It is possible to verify suppositions about the variances among the means of a variable (response variable) according to management with more than two categorical levels through the statistical analysis realized by ANOVA. The ANOVA software statistically tested the effect of the impact of the level of satisfaction factor associated with the environmental comfort of high schools in Juiz de Fora

and Macapá. It was tested if the value of the factor (Fcalculated) supplied by the statistical study is superior to the factor (Ftabulated).

## 4.1. High School in Juiz de Fora

The city of Juiz de Fora is in the southeast region of Brazil, in the state of Minas Gerais, in the Zona da Mata of the state, and the area of the city is 1.436 km$^2$, with an estimated population of 564.310 in 2018 [25]. It is located 255 km from Belo Horizonte (Capital of the state of Minas Gerais) and 183 km from the city of Rio de Janeiro (Capital of the state of Rio de Janeiro). The climate is tropical, warm and humid, with the average annual temperature of 20.6 ºC [25]. The two schools evaluated in this research in Juiz de Fora, Minas Gerais, specifically the Colégio Santa Catarina (Santa Catarina School) and Colégio Cristo Redentor (Cristo Redentor School), are the most traditional schools in the city [4,26].

The Colégio Santa Catarina was constructed in 1900 by the German Sisters, who dedicated themselves to the education of the children of the German Colony. Farmers and industrialists built the Colégio Cristo Redentor (Academia de Comércio) in March 1891, and it was the first institute of superior education of commerce in Brazil [26]. The questionnaires (Appendix A) were applied in the two high schools in Juiz de Fora (MG), on July 19, 2017. The number of students in the school is 1980, and 269 students in both high schools answered the survey related to comfort indicators, 57% from Colégio Santa Catarina (total of 956 students) and 43% from Colégio Cristo Redentor (total of 854 students) [26]. Through the sample calculation made by the Comentto calculator, considering the total number of 1980 students from the two schools, 5% sampling error, 90% confidence level, and with a homogeneous population, the minimum value found for the sample was 160. The number of students interviewed was 269; therefore, the research is valid.

The characteristics related to the environmental comfort in the high school classrooms in Juiz de Fora are shown in Table 1. The results of the survey are demonstrated in Table 3, considering all the students interviewed.

**Table 1.** Aspects of the classrooms of the Santa Catarina and Cristo Redentor schools, Juiz de Fora [4].

| | |
|---|---|
| Temperature in Juiz de Fora | 11 to 14 °C |
| Humidity of the air | 81% |
| Temperature in the class | 14 to 18 °C |
| Thermal Comfort | Heating, Ventilation and Air Conditioning (HVAC) is on just in the summer season;Natural ventilation most of the time of the day;<br>No roof insulation (strong insulation);<br>Simple glasses in windows;<br>Windows are efficiently insulated;<br>Double insulation of external walls;<br>There is no heating system inside the classrooms, but the external wall maintains the temperature;<br>Cover material: the windows used are simple, without special sealing and the external walls are made of solid bricks. |
| Visual Comfort | The windows are large (L = 2 m and H = 1 m), favoring natural lighting;<br>There are skylights in darker rooms, which benefits natural lighting;<br>Artificial lighting is produced by fluorescent lamps;<br>The artificial and natural lighting are adequate. |
| Acoustic Comfort | No acoustic treatment;<br>Simple glasses in windows;<br>No proximity to automobile traffic lanes;<br>Reverberation time is 0.86 s, which is considered high, according to the NBR 10152 [27] (0.55 s to 0.60 s). The cover material in the classroom is smooth and hard;<br>The level of the sound outside is just 35 dB, therefore, it does not interfere with comfort. There is no relevant noise from rooms near to the class;<br>sound level inside is from 60 to 65 dB, which is higher than what is determined by the NBR 10152, [27] (from 40 to 50 dB). |
| Air Quality | There is natural air circulation just in the summer season, the windows are always closed to preserve HVAC. |
| Ergonomic Comfort | The size of tables and chairs is not flexible. The high schools use standard furniture, not ergonomically suited for all students. |

Table 2 demonstrates the level of satisfaction of the children concerning comfort in the learning spaces in the schools in Juiz de Fora. The statistical investigation among groups was done, specifically, using Microsoft Excel (ANOVA) to undertake the statistical analysis of the comfort level regarding the results of environmental comfort. This defined that the Fcalculated = 17.81 > Ftabulated was equal to 3.09, probability was equal to 0.05, i.e., there was major variability among the information. The students' contentment about air quality, visual and thermal comfort was high, but the approval was inferior concerning acoustic comfort and was inadequate about ergonomic comfort [4].

**Table 2.** Results of the percentages of the responses given by the students of the high schools in Juiz de Fora, Minas Gerais, concerning environmental comfort [4].

| Environmental Comfort | Very Uncomfortable | Uncomfortable | Slightly Uncomfortable | Comfortable |
|---|---|---|---|---|
| Thermal | 2% | 9% | 14% | 75% |
| Visual | 1% | 6% | 12% | 80% |
| Acoustic | 10% | 13% | 39% | 38% |
| Air quality | 1% | 20% | 1% | 79% |
| Ergonomic | 14% | 28% | 29% | 29% |
| General | 1% | 11% | 30% | 58% |

### 4.2. High School in Macapá

The city of Macapá is located in the state of Amapá, in the northern region of Brazil, the Amazon region located on the equator line (Latitude 0°). The distance between Macapá and Belém (Capital of Pará) is 331 km, which can be traveled by boat, taking around 24 hours, or by plane, 1 hour. Access by land does not exist as the two cities are cut by the Amazon River [28]. The state of Amapá is separated from the other states of Brazil by the Amazon River, which hinders the entry of all types of products, interfering with the development of the city [29].

The estimated population, in 2018, was 493,634 and the area of the city is 6.564 km$^2$ [25]. The climate of the region is humid equatorial, divided into only 2 seasons: winter, occurring from December to May, with an average humidity of 80% and temperatures between 22 and 36 °C; and summer, the dry season, from June to November with an average humidity of 70% and temperatures between 26 and 40 °C [29].

The surveys on environmental comfort were conducted in two traditional schools in the city of Macapá (Amapá), namely the Tiradentes School and the Gabriel Almeida Café School, both of which were founded in the 1970s. The interviews were applied to 271 high school students, 54% of them from the Tiradentes School (total of 648 students) and 46% of the students from the Gabriel Almeida Café School (total of 479students). Through the sample calculation made by the Comentto calculator, considering the total number of 1127 students from the two schools, 5% sampling error, 90% confidence level, and with homogeneous population, the minimum value found for the sample was 151. The number of students interviewed was 271 and, consequently, the research is valid.

The date of the interview was February 2019, and this period in the city of Macapá is characterized as being winter; the humidity was 78% and the average temperatures ranged from 27 to 33 °C [29]. The aspects related to environmental comfort in the classroom in the high schools in Macapá are shown in Table 3. The results of the questionnaire (Appendix A) are presented in Table 4, considering all the interviewed students.

**Table 3.** Aspects of the classrooms of the Tiradentes and Gabriel Almeida Café schools, Macapá (AP) [28].

| | |
|---|---|
| Temperature in Macapá | 27 °C to 33 °C |
| Humidity of the air | 78% |
| Temperature in the class | 16 °C to 30 °C |
| Thermal Comfort | HVAC is usually on in every room every day. In several classrooms, the air conditioners or the windows are damaged;<br>No natural ventilation at any time of day; except when the air conditioner is damaged;<br>No roof insulation (strong insulation);<br>Simple glasses in windows;<br>Windows are not efficiently insulated;<br>No insulation of external walls;<br>Standard bricks is the material of the external walls. The windows are simple, without special sealing. |
| Visual Comfort | One-meter high glass windows throughout the exterior walls, which is not sufficient;<br>Artificial lighting is produced by fluorescent lamps;<br>The lamps are less than what would be necessary, with great distance between them;<br>The artificial and natural lighting are not adequate. |
| Acoustic Comfort | No acoustic treatment;<br>Simple glasses in windows;<br>Proximity to automobile traffic lanes;<br>Reverberation time is 0.77 s, which is considered high according to the NBR 10152 (0.55 s to 0.60 s). The cover material of the floor is ceramic or wood, there are a lot of glass windows and the tables and chairs are made by hard plastic;<br>Sound level outside: 60 dB, has interference in the learning places, and there is noise coming from the rooms near the classrooms;The level of the sound inside is 64 to 68 dB, above what is determined by the NBR 10152, according to which the acoustic comfort inside the classrooms has to be between 40 to 50 dB. |
| Air Quality | There is little natural air circulation as the windows are always closed to preserve HVAC. |
| Ergonomic Comfort | The size of tables and chairs is not flexible; the high schools use standard furniture, not ergonomically suited for all students. |

**Table 4.** Results of the percentages of the responses given by the students of high schools in Macapá [28].

| Environmental Comfort | Very Uncomfortable | Uncomfortable | Slightly Uncomfortable | Comfortable |
|---|---|---|---|---|
| Thermal | 17% | 16% | 18% | 49% |
| Visual | 19% | 22% | 18% | 41% |
| Acoustic | 9% | 15% | 34% | 42% |
| Air quality | 1% | 14% | 1% | 84% |
| Ergonomic | 8% | 17% | 26% | 50% |
| General | 1% | 20% | 32% | 47% |

Table 4 demonstrates the level of approval of the students regarding comfort in the learning places in the high schools in Macapá. The statistical study among groups was performed using Microsoft Excel (ANOVA). This software was used to undertake the statistical analysis of the comfort level regarding the results of environmental comfort. This defined that the Fcalculated = 13.51 > Ftabulated = 3.28 (probability equal to 0.05), i.e., there was relevant variability among the data. The students' approval was deficient regarding ergonomic comfort, however, it was very high regarding air quality comfort, and the approval was somewhat lower concerning acoustic, thermal and visual comfort.

Through the analysis of the information, it can be concluded that there is a strong need to provide assistance to schools, giving a main concern to thermal, visual, acoustic and ergonomic comfort, without forgetting the air quality indicators.

## 5. Results

Table 5 and Figure 1 demonstrate the similarities and differences among the percentages of environmental comfort sensed by the students of the schools of Macapá (Amapá) and Juiz de Fora (Minas Gerais).

**Table 5.** Parallel of percentages of the responses given by the students of the high schools of Macapá (MA) and Juiz de Fora (JF) concerning environmental comfort.

| Environmental Comfort | Very Uncomfortable | | Uncomfortable | | Slightly Uncomfortable | | Comfortable | |
|---|---|---|---|---|---|---|---|---|
| | **(JF)** | **(MA)** | **(JF)** | **(MA)** | **(JF)** | **(MA)** | **(JF)** | **(MA)** |
| Thermal | 2% | 17% | 9% | 16% | 14% | 18% | 75% | 49% |
| Visual | 1% | 19% | 6% | 22% | 12% | 18% | 80% | 41% |
| Acoustic | 10% | 9% | 13% | 15% | 39% | 34% | 38% | 42% |
| Air quality | 1% | 1% | 20% | 14% | 1% | 1% | 79% | 84% |
| Ergonomic | 14% | 8% | 28% | 17% | 29% | 26% | 29% | 50% |
| General | 1% | 1% | 11% | 20% | 30% | 32% | 58% | 47% |

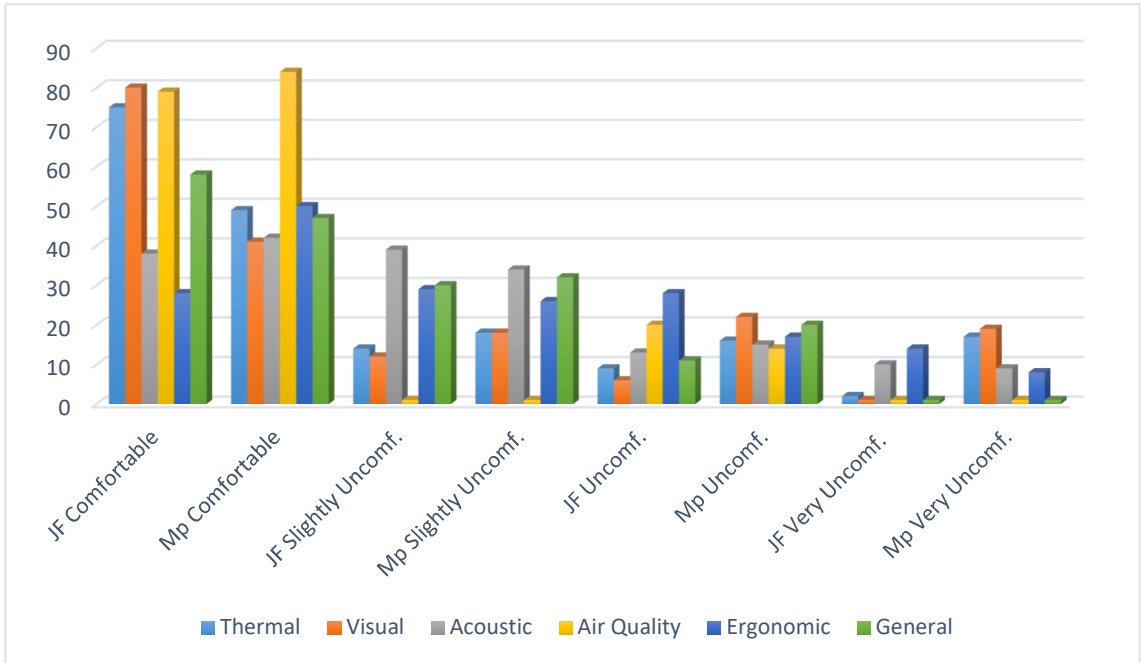

**Figure 1.** Percentage of the student satisfaction level related to environmental comfort in the high schools of Macapá (Mp) and Juiz de Fora (JF).

The statistical study among groups (Table 5), using Microsoft Excel (ANOVA), regarding the results of the environmental comfort for the high schools in Minas Gerais and Amapá showed that:

(i)     For the cities: Fcalculated = 0.003 < Ftabulated = 2.48 (probability equal to 0.05), i.e., there is a substantial variability among the cities. The students from Amapá (AP) and Juiz de Fora (MG) have different parameters regarding school buildings.

(ii)    For the environmental comfort: Fcalculated = 13.41 < Ftabulated = 2.28 (probability equal to 0.05), i.e., there is major variability of environmental comfort among the cities. Fifty-three percent of the students feel comfortable inside the learning spaces, which is an insufficient result.

As a result of the analysis of the data, it can be concluded that it is important to support the schools, giving preference to acoustic and ergonomic comfort.

Table 5 and Figure 1 demonstrated the general results of environmental comfort assessments are completely different, since Juiz de Fora (Minas Gerais) and Macapá (Amapá) have distinct aspects, such as flora, culture, climate and locations.

The results regarding general comfort (see Table 5) indicate that 58% of students from Minas Gerais and 47% of the students from Amapá are comfortable.

Only 38% of the students in the schools of Juiz de Fora (MG) and 42% in the schools of Macapá (AP) are satisfied regarding acoustics. That occurs because, in Minas Gerais, the sound level inside the

classroom is 60 to 65 dB and the reverberation time is 0.86 s, whereas in Amapá, the sound level is 64 to 68 dB, and the reverberation time is 0.77s. These results are superior to what is determined by the NBR 10152 [27], 40 to 50 dB inside the classroom, and reverberation time between 0.55 and 0.60 s. These high results occur due the fact that the walls that divide one room from another do not have adequate acoustic insulation, and, furthermore, the treatment of the materials of the classroom surface is hard and smooth, accentuating reverberation.

The level of ergonomic comfort presents low results (35%), in comparison with other results regarding environmental comfort, since there are standard chairs and tables in all schools, causing learning problems, discomfort and health problems in the students. Different sizes of furniture should be used among teenagers that may fit the biotype of each student.

## 6. Discussion

Students from Juiz de Fora and Macapá are comfortable with the air quality (79% and 80%). The students from Juiz de Fora are comfortable with the thermal and visual aspects (75%). Only 49% of the students from Macapá are satisfied with the thermal conditions, because sometimes the air conditioner is too cold or broken. Only 41% of the students from Macapá are satisfied with the lighting in the classroom, since the windows are not enough to let in an adequate amount of daylight, and the lamps are not of sufficient quality and efficiency. The general comfort in the city of Macapá presents a percentage of 47%, and in Juiz de Fora shows a percentage of 58%.

The comparison between Juiz de Fora (Southeast Region) and Macapá (Amazon Region) shows that, despite being in the same country, the two cities present a much larger difference in results (58% and 47%). This demonstrates a major need to elaborate or adapt methodologies and certifications for sustainability in school buildings so that they are specific for each of the regions of Brazil.

## 7. Conclusions

Brazil is a country with large dimensions, with a great cultural, economic and climatic diversity. Therefore, it is necessary to have several specific sustainability assessments for each region.

This study highlights the significance of comfort indicators for sustainability assessment for school buildings, since students stay around 25% of their day inside the learning place and environmental comfort affects the learning, health and concentration of the students.

The comparison of the indicators of environmental comfort, conducted in Juiz de Fora, in the southeastern region of Brazil, and in Macapá, in the Amazon region, intended to show the differences between the characteristics of the various Brazilian regions, making it necessary to elaborate specific criteria for sustainability evaluation methodologies for schools in each region.

The research has shown that these differences are very large by analyzing the levels of satisfaction of the students with regard to environmental comfort.

Increasing environmental impacts and improving people's quality of life are increasingly being sought, and sustainable buildings help achieve these objectives.

This work is relevant, as there is growing demand for sustainability in Brazil, with increasingly sustainable methodologies such as LEED, BREEAM, SBTool, AQUA methodologies being or intended to be applied in the country.

**Author Contributions:** Conceptualization, T.S.S., E.M.d.S. and M.A.; Methodology, T.S.S. Formal Analysis, E.M.d.S. T.S.S. and M.A; Investigation, E.M.d.S., T.S.S.; Writing-Original Draft Preparation, E.M.d.S. and T.S.S.; Writing-Review and Editing, M.A., T.S.S. and L.B.; Project Administration, T.S.S.

**Funding:** This research has not received any specific grants from public, commercial, and non-profit funding agencies.

**Acknowledgments:** The authors wish to thank Parque Escolar Company (EPE) for providing all material necessary for the execution of the article; and Abílio Ferreira, the managing director of Francisco de Holanda Secondary School, for showing the school and providing necessary information.

**Conflicts of Interest:** The authors declare that there is no conflict of interest.

**Appendix A**

This questionnaire was used by the author in her thesis, and in a previous article about the same issue, "Environmental comfort indicators for school buildings in sustainability assessment tools" [4], so it should remain the same.

Questionnaires—Environmental Comfort Indicators—Students
Grade Date:

1. Verify the sense of comfort where you are, regarding the temperature.

1.  Comfortable
2.  Very cold
3.  Cold
4.  A little cold
5.  A little hot
6.  Warm
7.  Very warm

2. Verify the sense of comfort, regarding the lighting.

1.  Comfortable
2.  A little uncomfortable with excessive lighting
3.  Uncomfortable with excessive lighting with lighting
4.  Very uncomfortable with excessive lighting
5.  A little uncomfortable with insufficient lighting
6.  Uncomfortable with insufficient lighting
7.  Very uncomfortable with insufficient lighting

3. Check the feeling of comfort, considering only the noise level.

1.  Comfortable
2.  Slightly noisy
3.  Noisy
4.  Very noisy

4. Verify the sense of comfort, regarding the level of air quality.

1.  Fresh
2.  Muffled
3.  Odorless
4.  Smelly
5.  Comfortable
6.  A little polluted
7.  Polluted
8.  Very Polluted

5. Verify the sense of comfort, regarding the level of ergonomic comfort, that is, the comfort offered by desks and the movement of students in the classroom.

1. Comfortable
2. A little uneasy
3. Uncomfortable
4. Very uncomfortable

6. Verify the sense of comfort regarding all the factors mentioned above.

1. Comfortable
2. Slightly uncomfortable
3. Uncomfortable
4. Very uncomfortable

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
