# Peer review of "Comparative Study of Comfort Indicators for School Constructions in Sustainability Methodologies: Schools in the Amazon and the Southeast Region of Brazil"

_sustainability, doi:10.3390/su11195216_

Round 1

Reviewer 1 Report

This is a clearly written paper on a relevant topic. The introduction is mainly sufficient , even if rather short, and methods and results are presented clearly. It is almost ready for publication, but there are a few points that I wish the authors would consider, to improve the paper.

First, what are the actual research questions of the study? Even though the paper in general is easy to follow, the purpose of the study is not well presented. This ought to be written already in the introduction.

Secondly, the discussion and conclusions could connect the findings to international context, along with the Brazilian context now used. The point of having different criteria for different regions is strong and relevant, so why not discuss the possibility to apply it internationally? Sustainability is a highly international research journal, so extending the discussion from this local case to international level would be recommendable and would increase the interest of readers. 

Thirdly, I wish to see a paragraph discussing the effects of the suggestion of regional criteria to environmental sustainability such as energy consumption of the school buildings. What kind of impact would it have?

I wish to thank the authors for an interesting paper and wait to see it published.

Reviewer 2 Report

To accept the paper for publication, the following changes ought to be made:

The form of the paper should change considerably so that it is not another version of the paper previously published in Sustainability (Saraiva, T., Almeida, M., Bragança, L., Barbosa, M.T. 2018. Environmental comfort indicators for school buildings in sustainability assessment tools. Sustainability v.10, p.1849, 2018; DOI: 10.3390/su10061849). The study’s aim and objectives as well as its rationale should be stated loudly and clearly in the Introduction section so that the readers know exactly what they are about to read. A separate section with a literature review must be written and added to the paper. Regarding the Methodology section, a description of the student population of each school as well as the sampling method followed to extract the sample must be added. Moreover, the authors should describe the method followed in terms of questionnaire completion and refer to the literature sources they used to design the questionnaire. Lines 288-329 involve study results and as such these lines should be transferred to the Results section. A Discussion section should also be added in which the findings of the specific study will be commented in relation to the findings of other studies.

Round 2

Reviewer 2 Report

All changes have been made and I suggest that the paper is accepted for publication